# Variations in Biodistribution and Acute Response of Differently Shaped Titania Nanoparticles in Healthy Rodents

**DOI:** 10.3390/nano13071174

**Published:** 2023-03-25

**Authors:** Martina B. Violatto, Giovanni Sitia, Laura Talamini, Annalisa Morelli, Ngoc Lan Tran, Qian Zhang, Atif Masood, Beatriz Pelaz, Indranath Chakraborty, Daxiang Cui, Wolfgang J. Parak, Mario Salmona, Neus G. Bastús, Victor Puntes, Paolo Bigini

**Affiliations:** 1Department of Molecular Biochemistry and Pharmacology, Istituto di Ricerche Farmacologiche Mario Negri IRCCS, Via Mario Negri 2, 20156 Milano, Italy; martina.violatto@marionegri.it (M.B.V.); laura.talamini@marionegri.it (L.T.); annalisa.morelli@marionegri.it (A.M.); mario.salmona@marionegri.it (M.S.); 2Experimental Hepatology Unit, Division of Immunology, Transplantation and Infectious Diseases, IRCCS San Raffaele Scientific Institute, Via Olgettina 58, 20132 Milano, Italy; sitia.giovanni@hsr.it (G.S.); tran.ngoclan@hrs.it (N.L.T.); 3Department of Instrument Science and Engineering, School of Electronic Information and Electrical Engineering, Institute of Nano Biomedicine and Engineering, Shanghai Jiao Tong University, 800 Dongchuan RD, Shanghai 200240, China; qianzhang0130@163.com (Q.Z.); dxcui@sjtu.edu.cn (D.C.); 4Karachi Institute of Radiotherapy and Nuclear Medicine (KIRAN), 75530 Karachi, Pakistan; atif_cancer@hotmail.com; 5Centro Singular de Investigación en Química Biolóxica e Materiais Moleculares (CiQUS), Departamento de Química Inorgánica, Universidade de Santiago de Compostela, 15782 Santiago de Compostela, Spain; beatriz.pelaz@usc.es; 6School of Nano Science and Technology, Indian Institute of Technology Kharagpur, Kharagpur 721302, India; chemistry.indra@alumni.iitm.ac.in; 7Center for Hybrid Nanostructures (CHyN), Universität Hamburg, Luruper Chaussee 149, 22607 Hamburg, Germany; wolfgang.parak@uni-hamburg.de; 8Vall d’Hebron Institut de Recerca (VHIR), 08035 Barcelona, Spain; neus.bastus@icn2.cat (N.G.B.); victor.puntes.icn@gmail.com (V.P.); 9Institució Catalana de Recerca i Estudis Avançats (ICREA), 08010 Barcelona, Spain; 10Institut Català de Nanociència i Nanotecnologia (ICN2), Consejo Superior de Investigaciones Científicas CSIC and the Barcelona Institute of Science and Technology (BIST), Campus UAB, Bellaterra, 08036 Barcelona, Spain; 11CIBER en Bioingeniería, Biomateriales y Nanomedicina, CIBER-BBN, 28029 Madrid, Spain

**Keywords:** titanium dioxide nanomaterial, physico-chemical properties, biodistribution, nanotoxicity

## Abstract

Titanium dioxide nanoparticles (TiO_2_ NPs) are one of the main sources of the nanoparticulate matter exposure to humans. Although several studies have demonstrated their potential toxic effects, the real nature of the correlation between NP properties and their interaction with biological targets is still far from being fully elucidated. Here, engineered TiO_2_ NPs with various geometries (bipyramids, plates, and rods) have been prepared, characterized and intravenously administered in healthy mice. Parameters such as biodistribution, accumulation, and toxicity have been assessed in the lungs and liver. Our data show that the organ accumulation of TiO_2_ NPs, measured by ICP-MS, is quite low, and this is only partially and transiently affected by the NP geometries. The long-lasting permanence is exclusively restricted to the lungs. Here, bipyramids and plates show a higher accumulation, and interestingly, rod-shaped NPs are the most toxic, leading to histopathological pulmonary alterations. In addition, they are also able to induce a transient increase in serum markers related to hepatocellular injury. These results indicate that rods, more than bipyramidal and spherical geometries, lead to a stronger and more severe biological effect. Overall, small physico-chemical differences can dramatically modify both accumulation and safety.

## 1. Introduction

The peculiar physico-chemical properties of engineered nanoparticles (NPs), such as composition, size, shape and surface features, make them attractive for use in the medical, agricultural, industrial and manufacturing fields [1]. These unique properties not only determine their utility for several applications, but also their toxicity [2]. Although hundreds of tons of NPs are produced and industrially processed, the connection between nanomaterial properties (in given exposure scenarios) and elementary biological responses (associated with pathological responses) are poorly known, and are used to signpost potential for hazard. In particular, knowledge of the biological interaction between different NPs derived from large-scale production processes and biological matrices (cells, tissues, whole animals) is a priority for defining their real impact on human health [3]. In the last decade, an increasing body of evidence has demonstrated the potential effect of NPs in respiratory and digestive systems, and their ability to enhance inflammatory responses in peripheral organs [4,5,6]. Thus, NP toxicology and the basis for their interaction with living matter became of great interest. Although the results arising from in silico and in vitro analyses are definitely pivotal to gain initial knowledge on NPs’ potential toxicity (depending on their geometry or surface charge), they are quite often hardly transferable to mammals [7]. The main reasons for this mismatch stem from the impossibility of recapitulating, either in cells or through specific algorithms, the complex and only partially known interplay resulting from exposure to the final destination (accumulation, transformations—such as protein corona—clearance, and toxicity) of the NPs in multicellular organisms. In a recent study, we demonstrated the strong influence of the geometry on the biodistribution of gold NPs [8]. Interestingly, neither size nor shape led to a relevant toxicity in treated mice. Similarly, changes in the surface and the shape of gold NPs strongly influenced the biodistribution but led to a minimal effect on the overall blood count and an almost complete absence of modifications in basal hematochemistry (creatinine, bilirubin, transaminases and uric acid) [9].

Gold NPs can be considered attractive for nanomedicine due to their theranostic properties. However, these results cannot so easily be translated to assess the relationship between geometry and in vivo effect in the field of nanosafety. Indeed, other materials are produced and dispersed by environmental pollution or industrial production. Among this broad range of materials, titanium dioxide NPs (TiO_2_ NPs) seem to play an even greater role in nanosafety. Due to their physico-chemical properties, TiO_2_ NPs are used as food additives (chewing gums, candies, sweets, and cheeses), high refractive index products (coatings, plastics, paints), photocatalyst agents (in water treatment and air purification), sunscreens, and toothpastes [10,11,12]. Titanium oxide is chemically stable and very difficult to dissolve. Therefore, its inertness and the persistence inside the body may represent a critical point concerning its safety, and large-scale production plans should pay great attention to this aspect.

Similar to other nanomaterials and chemicals, its risk is strongly influenced by the mutual influence of a series of different factors, including the geometry of the NPs. In a recent study carried out in C. elegans (a nematode universally considered a reliable model of human pathophysiology [13]), we reported that rod-shaped NPs were the most toxic, greatly impairing pharyngeal function, reproduction and larval growth, whereas bipyramidal and spherical shapes were less toxic, even if no difference in terms of biodistribution and accumulation were found.

The route of exposure obviously affects the in vivo absorption, biodistribution, and toxicity of NPs. Although the main routes of entry of NPs are skin, airways or digestive tract, it is important to emphasize that a great source of risk is tightly regulated by their penetration into the bloodstream and their consequent overall distribution in many organs and cells [14]. To this end, the influence of the geometry of different synthesized TiO_2_ NPs (bipyramids, plates, and rods) and their fate in terms of biodistribution, clearance, toxicity were investigated after a single intravenous treatment in healthy immunocompetent mice.

## 2. Materials and Methods

### 2.1. NP Synthesis and Characterization

Shape-controlled synthesis of TiO_2_ NPs was achieved following a modified nonaqueous surfactant-assisted synthesis reported by Gordon et al. [15]. TiF_4_, TiCl_4_ and/or TTIP were used as Ti precursors, ODE (1-octadecene) as a solvent, OLA (oleic acid) as a surfactant, and OLAM (Oleylamine) or ODOL (1-octadecanol) as cosurfactants. The control of the final morphology is achieved by the choice of titanium precursor and the co-surfactant. All syntheses were performed using standard Schlenk line techniques under a nitrogen atmosphere.

*Chemicals.* Titanium(IV) fluoride (TiF_4_, 99%, Acros, Geel, Belgium #7783633), titanium(IV) chloride (TiCl_4_, 99%, Strem, Bischheim, France), #221150), titanium(IV) isopropoxide (TTIP, 97%) 1-octadecene (90%, Alfa Aesar, Kandel, Germany, #L11004) (1-ODE), 1-octadecanol (1-ODOL, 97%), oleylamine (OLAM, 70%, Acros, Geel, Belgium, #129541000), oleic acid (OLA, 90%, Sigma Aldrich, Milan, Italy, #O1008), and tetradecylphosphonic acid (TDPA,98%, Sigma Aldrich, Milan, Italy, #736414) were used as obtained from the supplier.

*Preparation of Ti stock solutions.* TiF_4_ stock solution consists of 0.2 M TiF_4_ and 1.0 M OLA in 1-ODE. TiCl_4_ stock solution consists of 0.2 M TiCl_4_ and 1.0 M OLA in 1-ODE. TTIP stock solution consists of 0.2 M TTIP and 1.0 M OLA in 1-ODE. The TiF_4_ stock solution is stirred on a hot plate set to 80 °C to promote the dissolution of TiF_4_. Once dissolved, the TiF_4_ stock solution is orange-brown, and the TiCl_4_ and TTIP stock solutions are dark brown.

*Synthesis of TiO_2_ NPs (bipyramids).* Highly monodisperse anatase TiO_2_ tetragonal bipyramids (10 nm side length) were synthesized using a mixture of TiF_4_/TiCl_4_ precursors in the presence of OLAM.

*Synthesis of TiO_2_ NPs (rods).* Highly monodisperse brookite TiO_2_ rods (50 nm side length, 15 nm width) were synthesized using TTIP as a precursor in the presence of OLAM.

*Synthesis of TiO_2_ NPs (plates).* Highly monodisperse TiO_2_ plates (50 nm side length) were synthesized using a mixture of TiF_4_/TiCl_4_ precursors in the presence of 1-ODOL.

*General Synthetic Procedure.* A general procedure for the preparation of TiO_2_ NPs proceeds as follows. In a 125 mL flask, 30 mmol of cosurfactant (OLAM or 1-ODOL), 10.2 mL of 1-ODE, and 0.48 mL (1.5 mmol) of OLA are combined and degassed at 120 °C for 1 h. For a 1:1 mixture, TiF_4_ and TiCl_4_ stock solutions are mixed at equal volume in the glovebox. After degassing the flask, 1.5 mL of the desired stock solution (or mixed stock solution) is added at 60 °C. Then, the solution is quickly heated to 290 °C and held at this temperature for 10 min to allow for the formation of seed NPs. An 8 mL portion of the chosen stock solution is then pumped into the flask, which is kept at 290 °C, with a rate of 0.3 mL·min^–1^, using a syringe pump. Afterward, the heating mantle is removed, and the flask is left to cool naturally to ambient temperature. After the synthesis, the reaction contents are first diluted with a small volume of toluene and centrifuged at 6000 rpm to separate the NPs by precipitation. The NPs are then re-dispersed through the addition of toluene and 100 μL of OLAM with sonication. Insoluble surfactant and agglomerated NPs are removed through centrifugation (i.e., the precipitate is discarded and the supernatant containing the dispersed NPs is kept). A mixture of 2-propanol and methanol is added to precipitate the NPs, and centrifugation at 6000 rpm is used to precipitate the NPs before exchanging surfactants and redispersing them in water (see next section). This washing process was repeated twice.

*Ligand exchange of the TiO_2_ NPs.* The OLA/OLAM/ODOL ligands present in the TiO_2_ NPs were exchanged with an alkylphosphonic acid via a standard ligand exchange process to further enhance their colloidal stabilities [16]. Firstly, the OLA/OLAM/ODOL coated TiO_2_ NPs, as dispersed in toluene, were precipitated with methanol/ethanol by centrifugation (1000 rpm, 5 min), followed by discarding of the supernatant and redissolution of the NPs in CHCl_3_. This washing step was repeated until the TiO_2_ NPs could no longer be redispersed in CHCl_3_ (at least six times). In this case, one can consider that the TiO_2_ NPs had lost solubility through the removal of most of the original capping ligands from their surface. Then, 2 mL of CHCl_3_ was added to the NP precipitate under vigorous magnetic stirring at room temperature, and 1 mL of TDPA 0.2 M solution in CHCl_3_ was added, dropwise, until the solution turned transparent. The sample was continually stirred overnight. The next day, the NPs were precipitated by methanol and centrifugation (1000 rpm, 5 min), and the supernatant was discarded. After re-dispersion of the NPs in CHCl_3_, the washing process was repeated again by adding methanol, centrifugation, and redissolution in CHCl_3_.

*Phase transfer of TDPA coated TiO_2_ NPs to water by overcoating with an amphiphilic polymer.* The amphiphilic polymer poly(isobutylene–alt–maleic anhydride)–graft–dodecyl (PMA; 0.5 M monomer unit concentration in CHCl_3_) [16,17] was added into NPs; the solution of TDPA coated TiO_2_ NPs in CHCl_3_, and more CHCl_3_ was added to dilute the mixture according to previously published protocols [16,17,18]. The ratio of added PMA monomer units per effective NP surface (R_p/area_) was optimized experimentally. In the case of the platelet-shape TiO_2_ NPs, for example, ratio R_p/area_ = 600 nm^−2^ was used. Then, the solution was placed in the rotavapor at 60 °C for 10 min, the temperature was increased to 70 °C at 950 mbar, and the solvent (i.e., chloroform) was slowly evaporated until the sample was completely dried. The resulting solid NP film in the flask was redissolved under vigorous stirring in SBB12 buffer (sodium borate buffer, 50 mM, pH 12), until the solution turned clear [16]. Finally, the sample was concentrated with 100 kDa ultrafiltration filters (Sartorius Stedim, Goettingen, Germany) by centrifugation. The eventual excess of polymer micelles was removed through agarose gel electrophoresis [19]. In a final step, the solvent was exchanged to Milli-Q water by ultrafiltration. Dynamic light scattering (DLS) was used as a control to verify that the polymer-coated TiO_2_ NPs were well dispersed [16].

### 2.2. Animals and Treatments

The “Mario Negri” Institute for Pharmacological Research IRCCS adheres to the principles set out in the following laws, regulations, and policies governing the care and use of laboratory animals: Italian Governing Law (D.lgs 26/2014; Authorisation n.19/2008-A issued 6 March 2008 by Ministry of Health); Mario Negri Institutional Regulations and Policies providing internal authorization for persons conducting animal experiments (Quality Management System Certificate, UNI EN ISO 9001:2015, Reg. No. 6121); the NIH Guide for the Care and Use of Laboratory Animals (2011 edition), and EU directives and guidelines (EEC Council Directive 2010/63/UE). This work was reviewed by the IRCCS-IRFMN Animal Care and Use Committee (IACUC) and then approved by the Italian “Istituto Superiore di Sanità” (code: 42/2016-PR).

Eight-week-old male CD1 mice were housed in ‘specific pathogen-free’ animal rooms at a constant temperature of 21 ± 1 °C, humidity 55 ± 10%, with a 12 h light/dark cycle and ad libitum access to food and water. Mice were randomly divided into three groups receiving bipyramids, plates, and rods, respectively (n = 9 for each experimental group). Briefly, all animals received the same dose of each NPs preparation (6 mg/kg), diluted in 200 μL of injection grade distilled water, by intravenous injection. At the selected time points (pre injection, 1, 12, 24, 96, and 168 h), mice were anesthetized and blood was taken by retro orbital bleeding for complete blood counts, and the serum analyzed for markers of toxicity. Furthermore, at 1, 24, and 168 h after NPs injection, three mice for each group were killed and their organs collected for histological analysis.

### 2.3. ICP-MS Sample Preparation and %ID Calculation

Inductively coupled plasma mass spectrometry (ICP-MS) analysis, in which the amount of elemental Ti was determined, was carried out following previously published protocols for other NP materials [8,17,20,21]. The samples were first digested by the addition of 3 mL of ultra-pure (67 wt%) HNO_3_ (Fisher Chemical, Milan, Italy) under constant agitation in 50 mL falcon tubes for 72 h at 22°C, until the solution became clear and no organics (i.e., rests of tissue) were left in the tube. Some 100 µL of these digested samples was taken and further digested with 100 µL of HF acid for 48 h in order to digest the TiO_2_ NPs. Before ICP-MS measurements, the samples were diluted additionally by a factor of 10, using a low matrix consisting of 2 wt% HCl. This actually enhances the detection limit as well as reducing the background over all samples, and it ensures the protection of the machinery from corrosive HF. All vials and working materials had been either cleaned using freshly prepared aqua regia for 2 h, followed by boiling in Milli-Q water, or were sterile and clean non-reusable consumables. The samples were introduced into the ICP-MS setup through a perfluoroalkoxy-alkane (PFA)-based microflow spray chamber, wherein the aqueous sample was nebulized, introduced into the argon gas flow, and transported to the torch, where it was ionized in an argon plasma of around 6000 °C. After ionization, the sample was presorted using an omega lens, separated element-wise in a quadrupole field through the mass to charge rate, sorted again using kinetic barriers and a charged lens system, and finally detected with either an analog or a digital detector depending on the count rate. The percent of injected dose (%ID) was calculated as the percentage of Ti found in one organ referring to the total amount of injected Ti [17].

### 2.4. Blood Analysis and Histopathology

The extent of hepatocellular injury and toxicity was monitored by measuring serum alanine aminotransferase (sALT). Markers of cell toxicity such as serum aspartate aminotransferase (sAST) and lactate dehydrogenase (LDH) activity were measured at the indicated time points after NP injection. sALT, sAST and LDH activity were measured using a IFCC (International Federation of Clinical Chemistry and Laboratory Medicine) optimized kinetic UV method in an Aries chemical analyzer (Werfen Instrumentation Laboratory S.p.A., Milan, Italy) and expressed as U/L (Units/Liter). Complete cell counts were measured in whole blood collected in EDTA coated microvettes (Sarstedt, Nümbrecht, Germany) utilizing an automated cell counter (IDEXX Procyte Dx, IDEXX Laboratories).

Each analysis was validated by a certified biochemical chemistry and haematology specialist using quality control serums (CQI) or blood, in the San Raffaele Mouse Clinic (http://research.hsr.it/en/services/mouse-clinic/hematologic-testing.html, accessed on 21 October 2021).

At time of autopsy for each mouse, different organs were sampled, fixed in zinc-formalin, processed, embedded in paraffin, cut and stained with hematoxylin/eosin, or further processed for immunohistochemical analyses as previously described [22]. A Picrosirius Red (0.5 g Sirius Red F3B (C.I. 35782) + 500 mL picric acid solution—Sigma Aldrich, Milan, Italy) staining method was performed to reveal collagen deposition. Immunohistochemical staining was performed utilizing the following antibody: anti-F4/80 (clone A3-1, AbD Serotec, Milan, Italy). All images were acquired using an Aperio Scanscope CS2 system (Leica Biosystems, Milan, Italy) available at the SRSI Advanced Light and Electron Microscopy BioImaging Center (ALEMBIC). The images were identified as representative areas of interest within the total area of the specimen analyzed and exported as ImageScope snapshots.

### 2.5. Statistics

All the tests carried out for the physico-chemical characterization of the NPs were performed at least in triplicates. For in vivo experiments, the number of animals sacrificed at each data point was minimized (n = 3) according to the 3Rs principle. *p* values <0.05 were considered statistically significant and reported on graphs. Data are presented as mean values ± standard error of the mean (SEM) and analyzed using GraphPad Prism Software (version 7). The differences in Ti levels measured by ICP-MS were analyzed with a one-way ANOVA followed by a Bonferroni post hoc test (** *p* < 0.01, *** *p* < 0.001). For blood cell analysis, a one-way ANOVA followed by Dunn’s post hoc test was carried out, (* *p* < 0.05, ** *p* < 0.005), while for hepatic toxicity, an unpaired T-test was carried out (** *p* < 0.01).

## 3. Results

### 3.1. Synthesis and Characterization

For this study, we synthesized and characterized a set of TiO_2_ NPs. Figure 1, Figure 2 and Figure 3 report representative images of bipyramids, rods, and plates, respectively, acquired using scanning transmission electron microscopy (STEM).

The table below (Table 1) clearly reports TiO_2_ NP size. Measurements were performed counting at least 100 particles, and the average size and the standard distribution were obtained.

### 3.2. Influence of the “Shape” on TiO_2_ NP Biodistribution

To quantitatively determine the biodistribution of the NPs in the main organs, ICP-MS analysis of the tissue digests was performed to measure the elemental titanium concentrations. Figure 4 shows the obtained results reported as percentage of the injected dose (%ID) in mice sacrificed at 1, 24, and 168 h after a single administration of bipyramids, rods, and plates. Vehicle-treated mice (n = 3) were intravenously injected with normal saline and studied as an inner control to exclude a bias due to the natural presence of titanium in the body. As expected, no titanium was detected in all tissues of saline-administered mice (data not shown).

The content of TiO_2_ NPs in the whole blood is reported in Figure 4A. Because circulating immunocompetent cells may actively interact with NPs through cellular uptake and their efficiency can be influenced by the NP geometry, we decided to collect whole blood (serum and cellular components) instead of the exclusive evaluation of plasmatic levels. All NP shapes exhibit a short circulation time with a very rapid clearance and a modest uptake from blood components, with a concentration falling to 4%ID. Interestingly, a slight but detectable increase in NP content was observed over time in mice treated with rods and bipyramids, while the animals treated with plates showed a less marked and only transient interaction with blood cells.

Direct injection in the bloodstream leads to a fast and progressive penetration of NPs in different organs. Overall, the highest Ti concentration was found in the liver, followed by lung, kidney, spleen, and brain. As expected, the liver plays the main role in the capture of NPs after intravenous administration; similarly to a recent study performed by our group [8], a shape-dependent effect was observed when comparing bipyramids to plates and rods (Figure 4B). Whereas at the first hour, no difference was found among the groups, 24 h after administration, animals receiving bipyramids showed a drastic increase in Ti content as compared to the other experimental groups. However, the strong decrease in Ti levels at 168 h after injection suggests that TiO_2_ NPs, independently of their geometry, do not lead to long-lasting accumulation in liver parenchyma.

Considering the %ID, the NP kinetic found in the kidney and spleen (Figure 4C,D) is very similar; TiO_2_ NPs rapidly reach these organs, but their presence is extremely transient, and independent of their shape, they rapidly disappeared at the latter time-points. It is likely that renal and splenic filtering through resident macrophage uptake does not occur in these experimental conditions.

As expected, the brain showed a drastic reduction in the levels of Ti from 1 to 24 h after administration (Figure 4E). This trend was similar in each shape, and may be determined by the sharp decrease in the NP levels in blood. It is therefore likely that the contribution measured at 1 h was almost exclusively related to the presence of blood in the brain vessels. The lack of NP accumulation in brain is confirmed by their levels measured in animals sacrificed at 24 and 168 h after NP injection.

The process of accumulation of TiO_2_ NPs in lungs is very interesting (Figure 4F). Both bipyramid and plate levels strongly increased from the 1st to the 7th day after treatment. This accumulation cannot be attributed to the effect of circulating NPs in the bloodstream. Rods showed a lower amount of Ti accumulation compared to the two other shapes. However, accumulation over the time is also noticeable for this type of NP. The reason for this particular penetration in lung parenchyma cannot be elucidated by our studies. However, it is widely known that the lung is one of the most irrigated organs in the body, and it is therefore possible that the major accumulation of particulate matter happens in this tissue. On the other hand, it is unlikely that an active passage through the pulmonary barrier occurs in this latter phase [23].

In vivo, the majority of the injected dose is cleared from the bloodstream by cells of the mononuclear phagocyte system, a network of immune cells located in organs such as the liver, spleen and bone marrow. Conversely, the accumulation of these TiO_2_ NPs in filter organs is manifold lower if compared to many other classes of soft and metallic NPs of similar geometry and size [8,24,25,26]. Moreover, they do not pass biological barriers such as the blood–brain barrier, but have a great ability to accumulate in the lung parenchyma. The influence of the shape is moderate but not negligible. An interesting aspect to underline is that the ability of a specific type of NP to penetrate and accumulate is greatly influenced by the target organ. However, on the whole, bipyramids demonstrate greater accumulation than plates and rods.

### 3.3. Influence of the “Shape” on TiO_2_ NP Biological Effect

The main components of whole blood are blood cells and plasma. Blood cells include red blood cells, white blood cells (among which lymphocytes, neutrophils, and monocytes) and platelets. In Figure 4A, we reported the accumulation of the three types of TiO_2_ NPs with different shape in the blood, while in Figure 5, the impact of this accumulation is evaluated by directly counting the white blood cells from the blood withdrawn 12 h after administration.

Complete blood cell analysis showed that the total number of circulating lymphocytes was not altered by NP injection (Figure 5A); a little and non-significant increase was observed in mice treated with bipyramids, with no specific biological relevance.

A trend of neutrophils increasing was instead observed for all types of NPs compared to vehicle-treated mice (Figure 5B). Although not statistically significant, this seems slightly influenced by the NPs’ shape.

In contrast to circulating lymphocytes and neutrophils, a marked increase in monocytes was observed in all three experimental groups treated with TiO_2_ NPs compared to vehicle mice (Figure 5C), and this is particularly evident after plate and rod administration. Monocytes are the circulating cells devoted to the removal and the phagocytosis of many exogenous molecules by an active uptake process occurring after the opsonisation of the target. The behavior of rods is somehow correlated with the levels of Ti in blood, wherein a clear increase in Ti content was measured in animals treated with NPs of this shape (Figure 4A). It is therefore possible to hypothesize that a rapid adaptation of the system consisting of an increase of monocytes is needed to remove the NPs from the circulation. The reason why rods lead to a stronger activation is unknown. It is likely that the contact with plasma proteins may somehow modify the interaction between nanomaterials and monocytes. Regarding the relationship between protein corona and NP shape, it is important to mention a previous study carried out by our group, which revealed that silver wires strongly affect alveolar epithelial cells, whereas spherical particles have no effect. This supports the hypothesis that shape is one of the important factors that determine particle toxicity [27].

In contrast to this overall trend of monocytes increasing, a significant decrease in circulating platelets was observed after TiO_2_ NP injection in comparison with vehicle-injected mice (Figure 5D). Similar to monocyte measurement, animals treated with rods and plates showed a marked alteration of platelets. To this end, it is reasonable to speculate that platelets, by recognizing and interacting with NPs with an irregular 3D structure, become activated and degraded in the circulation.

As shown in Figure 4B, the levels of Ti accumulation in the liver of treated mice were transient, with the peak of accumulation at 24 h after the administration of NPs. Importantly, a shape-dependent accumulation was observed, with bipyramids reaching highest levels than plates and rod-shaped NPs. To define if this accumulation was associated with liver toxicity, a time-course was carried out to determine the levels of both alanine aminotransferase (sALT, Figure 6A) and aspartate aminotransferase enzymes (sAST, Figure 6B). These two transaminase levels are commonly measured to make a first diagnostic evaluation of hepatocellular injury. sALT is a more specific indicator of hepatocyte cell death than sAST, which may be released in the circulation by the death of other cell types along with lactate dehydrogenase (LDH, Figure 6C).

The dashed lines indicate the upper value of normality of each variable. Higher levels of all these markers compared to physiological ones were measured 12 and 24 h following rod injection, while only the LDH value was found to be higher after 12 h following plate treatment. Notably, these levels rapidly returned to normal values in the next time points, indicating that the administration of TiO_2_ NPs generates only a transient injury with the NPs of plate and rod shape. However, this result should be strongly taken into consideration in cases of chronic exposure. Interestingly, no effect was induced by bipyramids for all the hepatic markers at all the time points (red bars).

To determine liver pathology associated with the increase of liver enzymes induced by TiO_2_ NPs, groups of mice were killed at 1, 24, and 168 h after NP injection, and their livers were analyzed by histopathological analysis. Since inflammation and activation of tissue macrophages are often tightly related, immunohistochemistry for F4/80 (a well-characterized membrane protein, extensively used as a marker for mature tissue macrophages and Kupffer cells) was carried out in consecutive sections previously stained with hematoxylin and eosin (H&E). As shown in Figure 7, two consecutive representative sections of liver, respectively processed for H&E or F4/80, are reported for each experimental condition and at each time point. Microgranulomas are depicted as red hatched lines, while micronecrotic areas are depicted with yellow-hatched lines.

As expected, no relevant hallmarks of inflammation, steatosis, and other tissue alterations were observed in pathogen-free immunocompetent mice treated with vehicle. Moreover, no difference between sections from vehicle- and bipyramid-treated mice was detected. Indeed, in accordance with the measurement of blood cells and liver enzymes (Figure 5 and Figure 6), the hepatic parenchyma from mice treated with bipyramids showed neither morphological alterations nor variations in the amount and the localization of macrophages.

Very interestingly, small microgranulomas can be detected since from first hour of treatment with plates; these are characterized by a clustering of macrophages inside the liver parenchyma (red circle) of treated mice. They are associated with a modification of the parenchyma itself, as shown by H&E staining. Histological alteration becomes even more pronounced in mice sacrificed 24 h after the treatment, where hallmarks of necrosis occur (yellow circle) with peripheral aggregates of F4/80 immunopositive cells. In the early phase, microgranulomas (red circle) were also observed in the liver of rod-treated mice. At 24 h after treatment, the relatively big lesions (yellow circle) appear similar to micro-necrotic areas or small infarcts. This implies that rods must have induced a vascular event (i.e., platelets and coagulation cascade activation with the occlusion of relatively large hepatic vessels). For both plates and rods, the perturbation effect was nearly limited to the first days and completely reversed along the first week after the administration of NPs. This transient effect is very interesting, because it seems that NP injection was able to produce an acute response but does not seem to trigger a chronic inflammatory state despite their persistence. Notably, the lack of relevant alterations in mice treated with bipyramids seemed to exclude the relationship between NP penetration and accumulation (higher and sharper, even if transient, compared to the other shapes, Figure 4B) and toxic effects on the liver parenchyma.

In addition, to assess the extent of fibrosis in liver, Sirius Red staining was performed to detect collagen distribution in the samples. No evidence of collagen overexpression was observed in all TiO_2_ NP-treated groups compared to vehicle-treated mice. The typical feature characterized by the presence of collagen around liver vessels was detected. On the contrary, interlobular collagen deposition (marker for fibrosis onset) was absent in all experimental groups (Figure 7).

ICP-MS measurements (Figure 4F) clearly demonstrate that there is also difference concerning their long-lasting accumulation in lungs between differently shaped NPs. To verify both if this accumulation can lead to a direct modification of the parenchyma and if there is a correlation between the levels and the toxicity of TiO_2_ NPs, histopathological analyses were carried out (Figure 8).

Figure 8A shows representative images of lung parenchyma processed with both H&E, Sirius Red and the F4/80 antibody from mice killed 1, 24, and 168 h after treatment. The graph in Figure 8B reports the measurement of the level of immunoreactivity in mice sacrificed at the last time point. From the first hour after treatment, a marked pulmonary tissue alteration was observed following the treatment with rods, whereas both plates and bipyramids did not seem to produce the same effect. However, all treatments tend to modify the tissue over time; the lower panel describing the last time point clearly reveals a well detectable difference between the parenchyma structure of vehicle-treated and TiO_2_ NP-treated mice. This gradual modification may be due the progressive accumulation in this organ. In contrast to what is observed in the liver, interstitial inflammation with recruitment of F4/80 positive cells was weaker, mainly pronounced 168 h after treatment, and mostly enhanced in animals receiving bypiramids and rods. While the lungs from mice treated with bipyramids show a pronounced interstitial immunoreactivity in the alveoli only after 168 h, rod-treated mice showed an earlier alteration that seemed to have a trend of attenuation over time. Similarly to what was found in liver, no features of fibrosis were observed in all experimental groups tested; collagen fibers were substantially associated exclusively with physiological structures (the basement membrane, alveolar wall and alveolar septa).

## 4. Discussion

The aim of the study was to evaluate the interaction of different TiO_2_ NP shapes and host tissues after systemic injection in healthy, immunocompetent and specific pathogen-free (SPF) mice. To deeply characterize this interaction, we focused our attention on well-defined parameters such as NP kinetics (penetration, accumulation, and clearance) and alterations in specific targets. It is well known that titanium dioxide is used in many products that have a potential impact through different entry routes (skin, airways, gastrointestinal tract) [28]; after absorption by capillaries and vessels, it can diffuse into many organs [29].

It is widely known that most industrial TiO_2_ NPs have an extremely heterogeneous nature in terms of geometry. We have recently demonstrated that gold NPs with different shapes had specific distribution, peculiar accumulation, and different kinetics [8]. However, they did not appear to produce toxic effects on the main filter organs (kidney, liver, lung, and spleen) at least up to 5 days after intravenous administration. Based on this study, we resumed the same experimental model and approximately the same range of time points from the treatment to the sacrifice. Our previous study and, more generally, the extensive characterization of the behavior of gold NPs recently published by Tsoi et al. [30], confirmed the strong tropism of hard materials towards the liver and their ability to be internalized in reticulum endothelial system (RES) cells. Quite surprisingly, compared to the gold NPs kinetics, in this work, ICP-MS measurements revealed a low uptake of TiO_2_ NPs from livers and a complete disappearance from the 1st to the 7th day after administration. This transient phenomenon seems to exclude an efficient uptake from liver macrophages. The difference in biodistribution could be related to the smaller size of these TiO_2_ NPs (around 10 nm) compared to the AuNPs (about 50 nm).

Reduced absorption capacity by liver components could suggest (1) a greater filtering efficiency of the kidneys or spleen, which are the other two main target organs of the largest percentage of intravenously injected nanomaterials [31,32], and/or (2) a very fast clearance and excretion that would support the possibility of a safe exploitation of these materials. The very low splenic and renal levels of Ti do not seem to confirm the first hypothesis, even in spite of the lack of TiO_2_ measurements from feces and urine. The whole blood %ID was quite low for each type of NP; however, this was not unexpected, and was similar to that observed in mice treated with gold NPs. Therefore, the long-lasting permanence of the TiO_2_ levels in the whole body cannot be justified with the hypothesis that circulating NPs remain in plasma for 7 days. It is therefore possible that they remain somehow entrapped in the endothelium to be then progressively released and captured by the lungs from the 1st to the 7th day. Unexpectedly, an almost shape-independent accumulation (excluding rods) in the lungs was observed after systemic administration. Another possible hypothesis to explain the late accumulation in lungs is the progressive uptake from circulating cells, their activation, and their homing toward pulmonary parenchyma. The efficient penetration of human mesenchymal stem cells loaded with polymeric NPs in the lung has been demonstrated [33,34]. A delayed cell-mediated transport of NPs uptaken from circulating cells is therefore the most likely assumption.

Overall, our data concerning the measurement of NPs in different organs confirm that (1) in this type of administration, the accumulation of TiO_2_ NPs is low compared to many other hard materials, and they can be rapidly cleared up by kidneys; (2) the shape seems to affect only partially and transiently the general trend of accumulation. Interestingly, the behavior of rods is similar to that observed after the administration of gold nanorods, and therefore it could be a peculiarity of this shape; (3) tissue macrophages do not seem to play a crucial role in filtering TiO_2_ NPs, and; (4) the only process of long-lasting accumulation could be exclusively attributed to the lungs. In several cases of liver and lung injuries, the immunomodulatory response and the alteration of transaminases can lead to necrosis and fibrotic rearrangements of the tissue. Although in our study, the animals receiving TiO_2_ NPs showed altered levels of transaminases, some macrophage activation and infiltrates, the nature of them seems to be transient and at least upon a single treatment does not suggest a relevant tissue damage.

Kinetics and the biodistribution are important parameters to be considered in studies of bio–nano interaction. However, similarly to toxicology and pharmacology, the concentration can be somehow unrelated to their overall effect. For this reason, a deep survey of the effect of TiO_2_ treatment in the blood, liver and lungs has been carried out. Since the blood is the first tissue interacting with NPs after intravenous administration, we have evaluated the direct effect on the white blood cell and platelet count 12 h after treatment. Very interestingly, a slight tendency toward an increased number of white cells was found in treated animals, and this was more marked with regard to monocytes. Notably, monocytes are circulating macrophages, and their primary functions are the neutralization and the degradation/clearance of a large plethora of non-self elements. Increasing monocytes after injection is therefore not a surprising situation in SPF immunocompetent mice. This tendency does not appear to be exclusively influenced by NP geometry, even if plates and rods seem to have a similar effect both on the monocytes’ increase and in the simultaneous circulating platelets’ reduction. A very broad range of stimuli can lead to drastic reduction in platelets. However, this result is extremely interesting in light of the recent overview of the NP effect on platelets’ activation and coagulation trigger [35]. More specifically, experiments with both anatase and rutile TiO_2_ NPs mainly suggest the aggregation of platelets, whereas our study seems to show an increase in number without particular morphological differences or aggregations. Regarding the physico-chemical parameters that induce such changes, it is important to emphasize that an increase in platelets, particularly after chronic exposure, can have strong health effects in subjects with cardiovascular risk [36].

Alterations in white blood cells and platelets are not the only effect of TiO_2_ NP treatment. Despite the low accumulation in the liver parenchyma, early histological changes associated with transient inflammatory reaction and granulomas formation were clearly observed in mice sacrificed 1 and 24 h after NP administration. Although rods seem to produce a greater effect compared to other geometries, no evidence of long-lasting effect was suggested by single administration by any of the shapes. It is possible to argue that the extremely fast effect observable at 1 h after injection is caused by a bystander effect, and it is not associated with NP penetration. To understand whether accumulation and histopathological alterations could affect liver function, we performed kinetics on circulating levels of transaminases in all experimental groups. A transient increase in sALT and sAST levels was almost exclusively observed in mice treated with rods, and to a lesser extent, in those treated with plates.

Another target of these NPs is the lung. It is interesting to see that compared to many other studies, a persistent bioaccumulation in the animals treated with TiO_2_ NPs appears to be greater in the lung than in other filter organs. This result is quite surprising but very interesting, and the interaction with the lung is also shown by the relevant histological changes and the activation of the macrophages seen in animals treated with TiO_2_ NPs.

We also need to explain the significance and relevance of “shape” in the context of our study. The three types of NPs used here in fact had clearly different shapes. However, apart from shape, the other physicochemical parameters of these three types of NPs were also different, namely the volume per NP and also the surface area per NP. All of these parameters can influence the interaction of the NPs with cells. In fact, results depend on which metrics are used to distinguish between the different types of NPs [37,38]. Unfortunately, it is highly complicated to vary only one parameter, such as shape, while keeping all other parameters such as volume, surface area, etc. constant. Here, large multi-dimensional NP libraries, in which all parameters are varied, may help. However, studies on such libraries show that it is often not possible to relate biological effects to just one single physicochemical property [17]. Thus, “shape” in our study refers to the fact that three types of different NPs were used, all of which had different shapes. Any observed biological effect, however, cannot be directly correlated to shape only, but rather refers to the type of NP.

## 5. Conclusions

Overall, this study demonstrates that TiO_2_ geometry can alter the accumulation and response of the host; however, this effect may have different kinetics, depending on the considered organ. The reversibility of all these the effects can be easily explained by the nature of the treatment (single administration), but their potential risk through chronic exposure cannot be neglected. Although many other experiments should be carried out, selecting specific ways of administering and performing chronic exposure, this analysis could be a step-by-step platform for assessing the guidelines related to the chemical and physical parameters of NPs. The obtained results can be exploited by the industry in order to limit those NPs that are more dangerous to human health.

## Figures and Tables

**Figure 1 nanomaterials-13-01174-f001:**
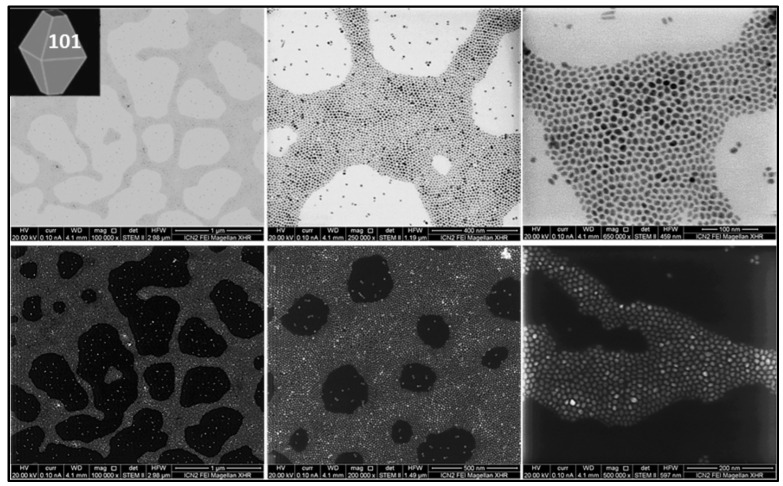
TiO_2_ bipyramids. STEM images of TiO_2_ NPs synthesized using TiF_4_ as a precursor in the presence of OLAM. In the presence of the co-surfactant OLAM, TiF_4_ produces highly uniform tetragonal bipyramidal anatase NPs (10 nm side length), which are significantly truncated perpendicular to the [001] direction.

**Figure 2 nanomaterials-13-01174-f002:**
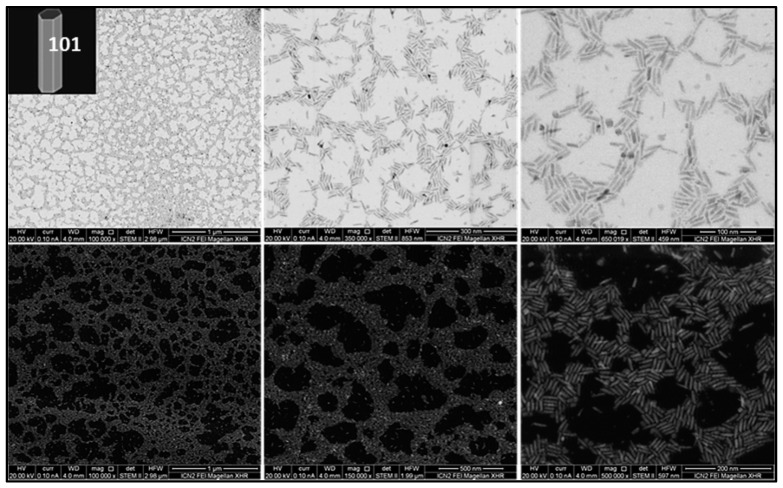
TiO_2_ rods. STEM images of TiO_2_ NPs synthesized using TTIP as precursor in the presence of OLAM. In the presence of the co-surfactant OLAM, pure phase brookite rods (50 nm side length, 15 nm width) are formed using TiCl_4_ as precursor.

**Figure 3 nanomaterials-13-01174-f003:**
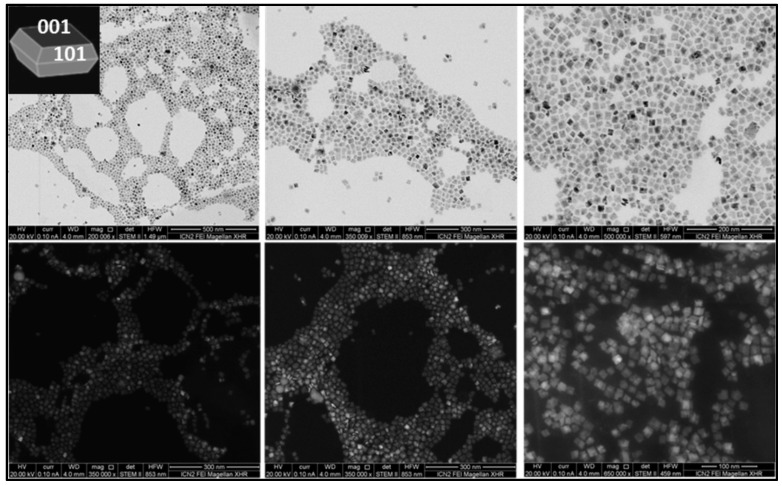
TiO_2_ plates. STEM images of TiO_2_ NPs synthesized using TiF_4_/TiCl_4_ as precursors in the presence of 1-ODOL. The use of the mixed TiF_4_/TiCl_4_ precursor in combination with 1-ODOL forms plates (50 nm side length, 10 nm width).

**Figure 4 nanomaterials-13-01174-f004:**
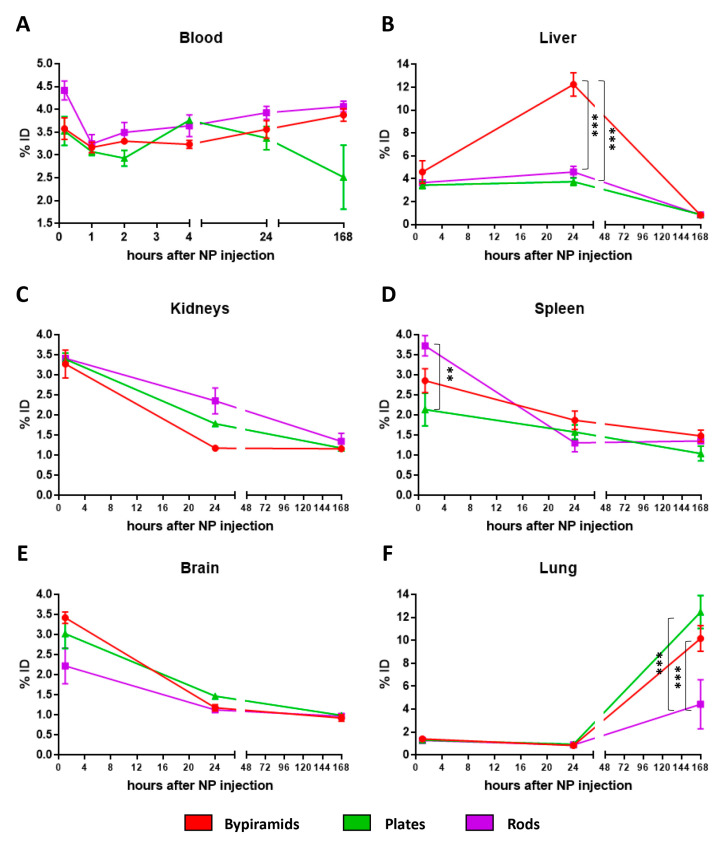
Ti organ content. Percentage of injected dose (%ID) measured by ICP-MS in blood derived from mice treated with different TiO_2_ NPs at 10′, 1, 2, 4, 24, and 168 h after treatment (**A**) and in the following collected organs ((**B**–**F**): liver, kidneys spleen, brain, and lung) derived from mice treated with different TiO_2_ NPs at 1, 24, and 168 h after treatment. The data are reported as mean ± SEM. A one-way ANOVA followed by a Bonferroni post hoc test was carried out. Significant difference (** *p* < 0.01, *** *p* < 0.001) when comparing the different shapes at the same time point is indicated.

**Figure 5 nanomaterials-13-01174-f005:**
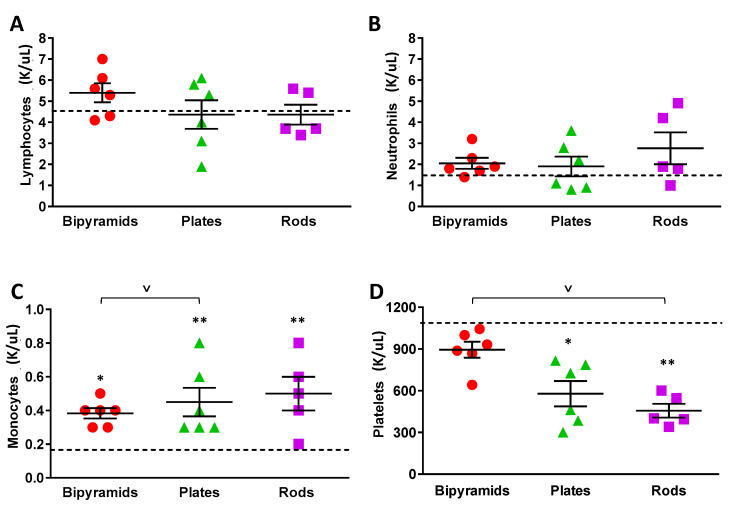
Complete blood cell analysis. Twelve hours after intravenous administration of TiO_2_ NP, six mice from each experimental group were phlebotomized, and blood was analyzed for total number of lymphocytes (**A**), neutrophils (**B**), monocytes (**C**), and platelets (**D**). Data are presented as mean ± SEM. A one-way ANOVA followed by Dunn’s post hoc test was carried out. This indicated significant difference (* *p* < 0.05, ** *p* < 0.005) between the NP-treated and vehicle-treated mice; (v *p* < 0.05) when comparing the different shapes. The means of each variable measured in control mice are depicted as hatched lines.

**Figure 6 nanomaterials-13-01174-f006:**
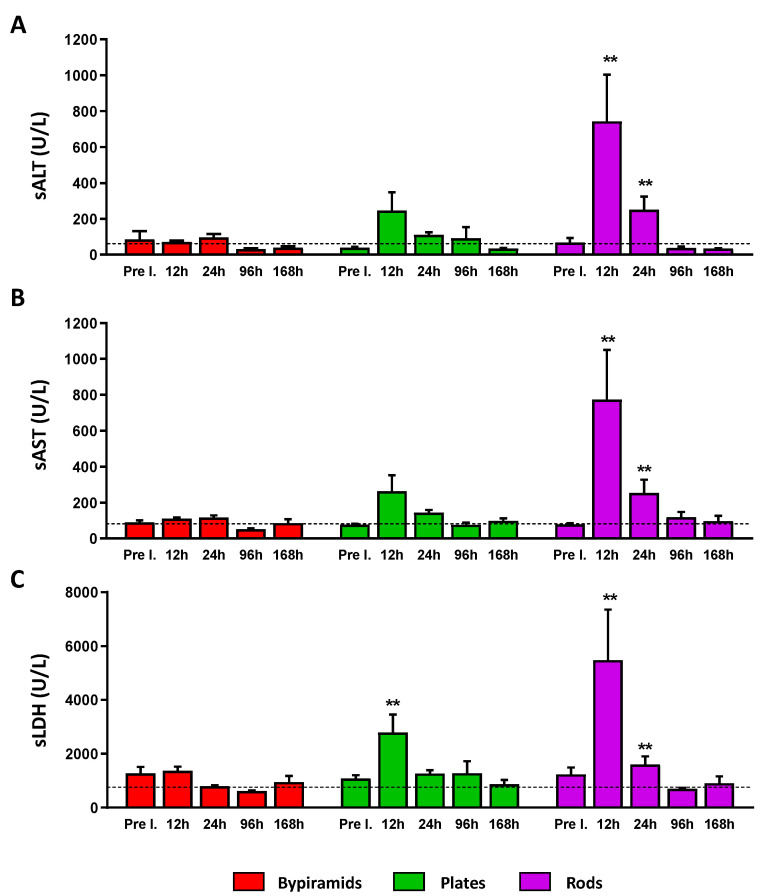
In vivo hepatic toxicity. Activity (Units/Liter [U/L]) of serum ALT (**A**), AST (**B**), and LDH (**C**) measured at the indicated time points in groups of mice injected with bipyramids, plates, and rods. The dashed lines indicate the upper value of normality of each variable (70 U/L for ALT, 83 U/L for AST, and 750 U/L for LDH). Pre I. = Pre injection. The values are expressed as mean ± SEM. An unpaired T-test was carried out; significant difference (** *p* < 0.01) when comparing NP-treated and vehicle-treated mice at the same time point is indicated.

**Figure 7 nanomaterials-13-01174-f007:**
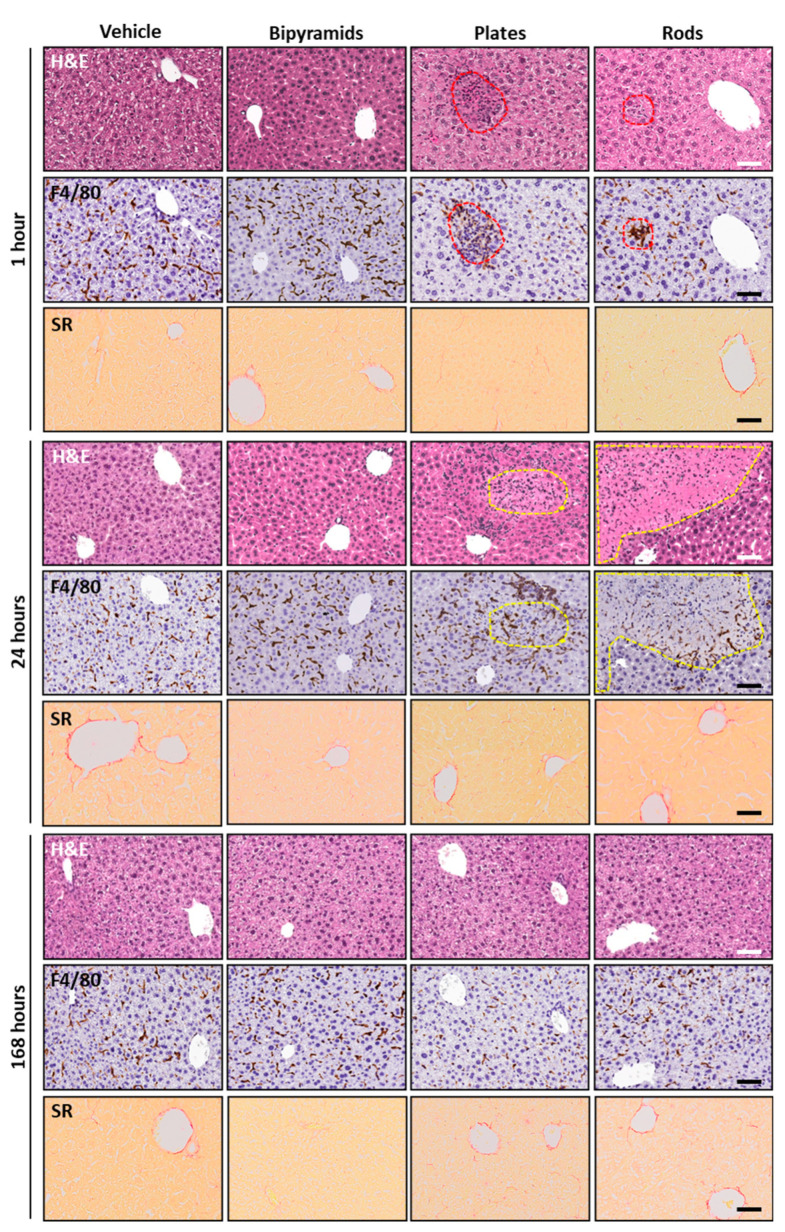
Liver histopathology and immunohistochemistry. Representative micrographs (H&E, upper panels, immunohistochemical staining, F4/80, middle panels and Sirius Red, lower panels) of the liver from mice injected with bipyramids, plates and rods, compared to vehicle-treated mice sacrificed at 1, 24 and 168 h after NP injection. Red hatched lines indicate microaggregation of F4/80 positive macrophages; yellow hatched lines indicate areas of hepatic infarcts. The scale bars represent 100 µm.

**Figure 8 nanomaterials-13-01174-f008:**
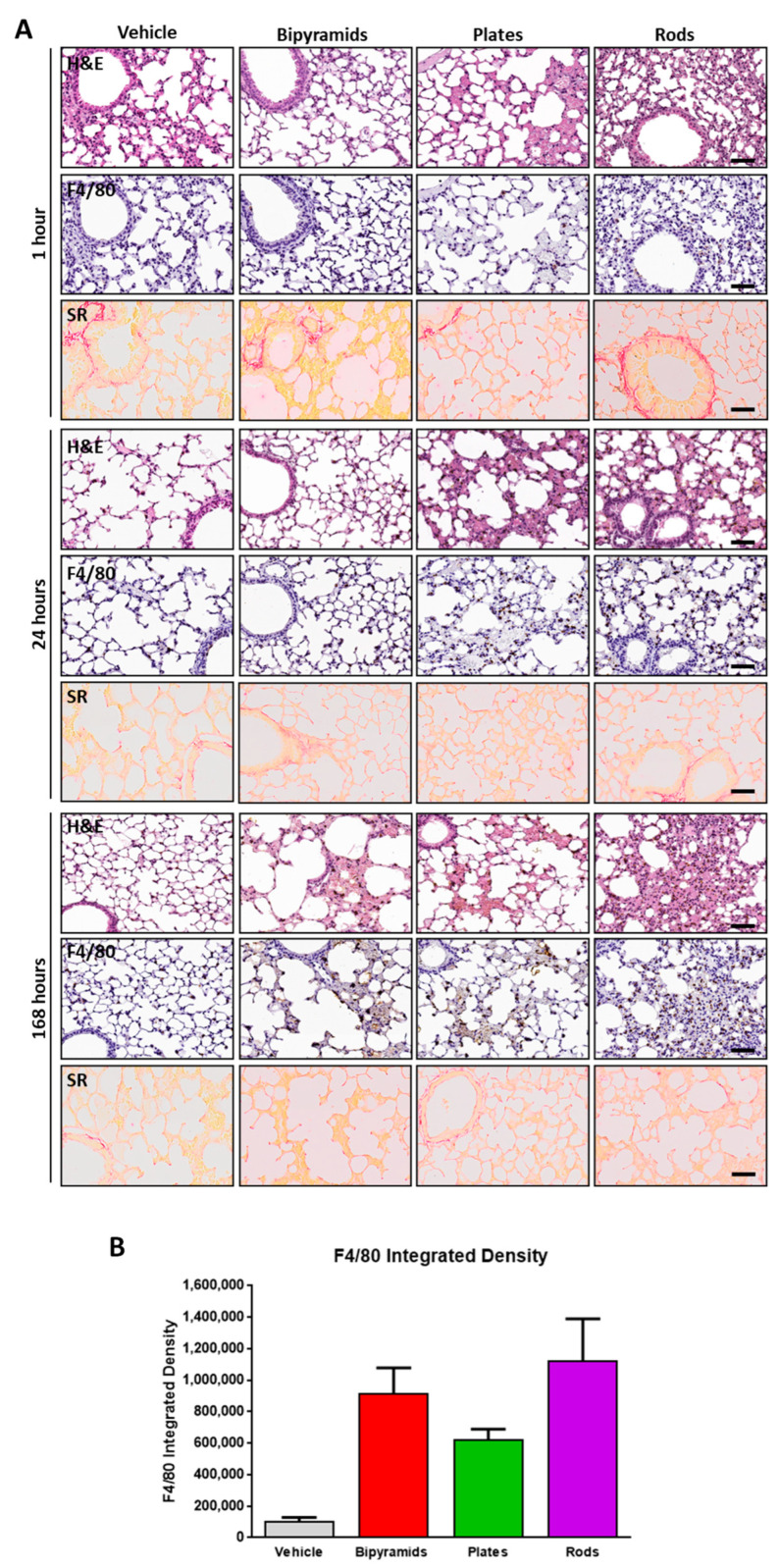
Lung histopathology and immunohistochemistry. (**A**) Representative micrographs (H&E, upper panels), immunohistochemical staining (F4/80, middle panels) and Sirius Red (lower panels) of the lungs of mice injected with bipyramids, plates, and rods, compared to vehicle-treated mice sacrificed at 1, 24, and 168 h after NP injection. The scale bars indicate 50 µm. (**B**) Histogram showing the spread of the pattern of immunoreactivity from sections of mice sacrificed 168 h after NP injection. The values were expressed as mean ± SEM.

**Table 1 nanomaterials-13-01174-t001:** Dimensional characterization of TiO_2_ NPs administered to mice.

Sample	Size Distribution
TiO_2_ bipyramids	7.5 ± 0.7 nm side
TiO_2_ rods	27.8 ± 2.1 nm length 6.3 ± 1.0 nm width
TiO_2_ plates	13.6 ± 2.1 nm side 4.5 ± 0.3 nm thickness

## Data Availability

The datasets presented in this study can be found in online repositories.

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
