# Peer review of "Variations in Biodistribution and Acute Response of Differently Shaped Titania Nanoparticles in Healthy Rodents"

_nanomaterials, 2023, doi:10.3390/nano13071174_

Round 1

Reviewer 1 Report

In this manuscript, authors engineered TiO2 NPs with various geometry (bipyramids, plates, and rods) have been prepared, characterized and intravenously administered in healthy mice. Parameters such as biodistribution, accumulation, and toxicity have been assessed in lungs and liver. The data in this study are through and are important to the filed. I suggest that this manuscript can be accepted after addressing the following minor issues.

1 the physiochemical characterizations of those three types of nanoparticles, including particle size and zeta potential, shall be summarized within one table to make them clear to readers.

2 I am curious that why the ID% at 0 time point is not 0 instead of around 2-5% in Figure 4. Authors should explain on that point or give the calculation methodology to avoid any misunderstanding to readers.

3 why is the histogram showing the spread of the pattern of immunoreactivity from liver sections of mice sacrificed 168 h after NP injection not shown in Figure 7.

Reviewer 2 Report

In this paper the authors synthesized  and characterized a set of TiO2  nanoparticles as bipyramids, rods, and plates, respectively. These nanoparticles were intravenously administered in healthy mice. Parameters such as biodistribution, accumulation, and toxicity were assessed in lungs and liver. All aspects were studied in detail and focused  attention on well-defined parameters such as NP kinetics (penetration, accumulation, and clearance) and alterations of specific targets.

The results showed that:

1) in this type of administration, the accumulation of TiO2 NPs and was rapidly cleared up by kidneys;

2) the shape affects only partially and transiently the general trend of accumulation. The behavior of rods is similar to that observed after the administration of gold nanorods and therefore it could be a peculiarity of this shape;

 3) tissue macrophages do not  play  a crucial role in filtering TiO2 NPs;

 4) the only process of long-lasting accumulation could be exclusively ascribable to the lungs.

In conclusion: this study demonstrated that TiO2 geometry can alter the accumulation and response of the host.

Some aspects of this study are unclear.

1. Why you used injection in blood of these nanoparticles? May be this is the lowest probability of these nanoparticles entering the body.

The protocol for obtaining these nanoparticles is were complicated and some aspect are not clear. For example why you use  “Phase transfer of TDPA coated TiO2 NPs to water by overcoating with an amphiphilic polymer”.
